# Tumor-associated reactive astrocytes aid the evolution of immunosuppressive environment in glioblastoma

Dieter Henrik Heiland [1,2,3,12], Vidhya M. Ravi[4,2,3,12], Simon P. Behringer[1,2,3], Jan Hendrik Frenking[1,2,3], Julian Wurm[1,2,3], Kevin Joseph [4,2,3], Nicklas W.C. Garrelfs[1,2,3], Jakob Strähle[2,3], Sabrina Heynckes[1,2,3], Jürgen Grauvogel[2,3], Pamela Franco[1,2,3], Irina Mader [5,6], Matthias Schneider[7], Anna-Laura Potthoff[7], Daniel Delev[8], Ulrich G. Hofmann [4,2,3], Christian Fung[2,3], Jürgen Beck[2,3], Roman Sankowski [9,3], Marco Prinz[9,10,11,3] & Oliver Schnell[1,2,3]

Reactive astrocytes evolve after brain injury, inflammatory and degenerative diseases, whereby they undergo transcriptomic re-programming. In malignant brain tumors, their function and crosstalk to other components of the environment is poorly understood. Here we report a distinct transcriptional phenotype of reactive astrocytes from glioblastoma linked to JAK/STAT pathway activation. Subsequently, we investigate the origin of astrocytic transformation by a microglia loss-of-function model in a human organotypic slice model with injected tumor cells. RNA-seq based gene expression analysis of astrocytes reveals a distinct astrocytic phenotype caused by the coexistence of microglia and astrocytes in the tumor environment, which leads to a large release of anti-inflammatory cytokines such as TGFβ, IL10 and G-CSF. Inhibition of the JAK/STAT pathway shifts the balance of pro- and anti-inflammatory cytokines towards a pro-inflammatory environment. The complex interaction of astrocytes and microglia cells promotes an immunosuppressive environment, suggesting that tumor-associated astrocytes contribute to anti-inflammatory responses.

[1] Translational NeuroOncology Research Group, Medical Center, University of Freiburg, 79106 Freiburg, Germany. [2] Department of Neurosurgery, Medical Center, University of Freiburg, 79106 Freiburg, Germany. [3] Faculty of Medicine, Freiburg University, 79106 Freiburg, Germany. [4] Neuroelectronic Systems, Medical Center, University of Freiburg, 79106 Freiburg, Germany. [5] Department of Neuroradiology, Medical Center, University of Freiburg, 79106 Freiburg, Germany. [6] Clinic for Neuropediatrics and Neurorehabilitation, Epilepsy Center for Children and Adolescents, Schön Klinik 83569 Vogtareuth, Germany. [7] Department of Neurosurgery, University of Bonn, 53113 Bonn, Germany. [8] Department of Neurosurgery, University of Aachen, 52062 Aachen, Germany. [9] Institute of Neuropathology, Medical Center, University of Freiburg, 79106 Freiburg, Germany. [10] Signalling Research Centres BIOSS and CIBSS, University of Freiburg, 79104 Freiburg, Germany. [11] Center for NeuroModulation (NeuroModul), University of Freiburg, 79106 Freiburg, Germany. [12]These authors contributed equally: Dieter Henrik Heiland, Vidhya M. Ravi. Correspondence and requests for materials should be addressed to D.H.H. (email: dieter.henrik.heiland@uniklinik-freiburg.de)

Glioblastoma multiforme (GBM) is the most common primary malignant brain tumor in adults, with an annual incidence of 3–4 cases per 100,000 people in Europe[1,2] and the United States[3]. In spite of the best available treatment, prognosis for patients with GBM is poor, with a median overall survival of merely 14–16 months[4–8]. In the last decade, numerous discoveries concerning the development, metabolism and alterations of gliomas were made. However, the interactions within the tumor microenvironment are poorly understood and require further research[9]. In malignant brain tumors, cellular components of the microenvironment occupy several functions either supporting the tumor growth or inhibiting its malignant properties[10]. Therefore, the focus of neuro-oncology research has moved to the cellular microenvironment of gliomas. Immune cells have been identified as the leading cell-type of the cellular environment; either migrating from other brain regions or from peripheral blood. The vast majority of immune cells were classified as macrophages/microglia (>95%), whereas dendritic cells (4.5%) were observed less frequently[11–13]. Darmanis et al., presented evidence that tumor-infiltrating macrophages and resident brain microglia preferentially occupy the tumor and peritumoral areas and directly affect the tumor through immune mediators[11]. Furthermore, myeloid cells were shown to enhance tumor growth, survival and dissemination via suppression of inflammation, promotion of angiogenesis and remodeling of the extracellular matrix[11]. In other non-neoplastic cell populations such as oligodendrocyte precursor cells (OPCs), neurons and mature oligodendrocytes, evidence of their supportive or inhibitory properties in glioblastoma is poor[11]. In the non-neoplastic brain, astrocytes are responsible for buffering the metabolic environment[14] and providing energy substrates for neurons. In traumatic, ischemic, inflammatory, degenerative and malignant diseases of the central nervous system (CNS)[15], reactive astrocytes revealed a considerable impact on the course of the disease. Recently, different reactive states of astrocytes were reported. On one hand, reactive astrocytes with A1-specific gene expression were first described after an inflammatory stimulus (lipopolysaccarid) and defined by an upregulation of pathways related to antigen presentation, complement activation and increased neurotoxicity[16,17]. On the other hand, alternative activated astrocytes also termed "A2-specific reactive astrocytes" occur under ischemic conditions, participate in scar formation and protect neurons and synapses by releasing neurotrophic factors and thrombospondins[16,17]. Reactive astrocytes within the tumor environment are poorly explored. Zhang et al. analyzed tumor-occupying astrocytes in three glioblastoma patients and uncovered similarities to astrocytes from fetal brains marked by increased proliferation[13]. In addition, Priego and colleagues explored a pro-metastatic program driven by STAT3 signaling in a subpopulation of reactive astrocytes surrounding metastatic lesions, modulating the immune surveillance[18]. The function of astrocytes in glioblastoma and their effect on tumor behavior and immune surveillance has not been investigated so far.

In our study, we identified a distinct transcriptional program in tumor-associated astrocytes in contrast to astrocytes purified from non-malignant specimens. Further, we showed that these tumor-associated astrocytes were marked by JAK/STAT pathway activation and CD274 expression, which were confirmed in a set of de-novo and recurrent glioblastoma specimens. Subsequently, our results indicate that a distinct reactive astrocytic subtype is mediated by a joint alteration of the surrounding secretome caused by both the tumor and microglia and that only in their presence, high concentrations of anti-inflammatory IL10 and TGFβ are detectable in the tumor microenvironment, which contributes substantially to the properties of an immunological "cold"-tumor environment.

## Results

**Human tumor-associated astrocytes revealed JAK/STAT activation.** We started our investigation by characterizing the transcriptional phenotype of reactive astrocytes in intraoperative human brain samples. Astrocytes were purified from the tumor core ($n = 7$), non-infiltrated brain regions ($n = 2$) or from entry cortex of epilepsy surgeries ($n = 3$). Non-infiltrated brain regions were defined by a minimum distance of 2 cm from the contrast-enhancing tumor core measured using intraoperative MRI navigation and confirmed using H&E staining (Supplementary Fig. 1a), A detailed description on the selection of patients and specimen preparation is given in the Supplementary Fig. 1a, b. Astrocytes were purified by immunoprecipitation from single cell suspensions ("immunopanning") as recently described by Zhang and colleagues[13] (Fig. 1a) using HepaCAM (Hepatic and Glial Cell Adhesion Molecule) as surface marker. Purified astrocytes were then characterized by RNA-seq based expression analysis. Further analysis on purification accuracy and data quality is given in the Supplementary Fig. 1c. In the reactive astrocytes from the tumor we identified an increased expression of immune associated genes such as *CHI3L1, HLA-DRA, CD274, HLA-DRB3, CD37* (Fig. 1b), as well as genes that contributed to proliferation (*MKI67, ANXA2*) and complement components (*C1S, C1R*) (Fig. 1b and Supplementary Fig. 2). In line with our findings, we identified an increased expression of *CHI3L1, HLA* related genes, *C1S, C1R,* and *MIK67* in tumor-associated astrocytes extracted from single-cell RNA-sequencing data (scRNAseq) released by Darmanis and colleagues[11], Supplementary Fig. 3. A gene set enrichment analysis revealed a significant increase of IFNγ-response and JAK/STAT pathway activation in tumor associated astrocytes (Fig. 1c, d).

**Tumor-associated astrocytes revealed anti-inflammatory state.** In recent years, various transcriptional subtypes of reactive astrocytes have been described. We hypothesized, that astrocytes, having multiple activation states, change their transcriptional profile along an axis between the "mature (mAC) - progenitor (APC)" state and/or along the "inflammatory-alternative activation" (Fig. 1e) We extracted the top 50 signature genes of the mature and progenitor stage of astrocytes[13], reactive astrocytes of the inflammatory (A1) and alternative (A2) subtype[17] and of astrocytes purified from hippocampal sclerosis specimens[13]. Tumor associated astrocytes from our dataset ($n = 7$), as well as astrocytes from external data ($n = 3$, Zhang and colleagues[13]) revealed a multidimensional shift towards a progenitor and *alternative activation* state (Fig. 1d). We additionally evaluated the scRNAsq data from Darmanis and colleagues[11] along our established classification axis, showing a similar transcriptional shift towards the progenitor state in one tumor-associated cluster (C2) and *alternative activation* state in the other tumor-associated cluster (C3), Supplementary Fig. 4. Further, we validated our novel marker genes *CD274* and *CHI3L1* along with STAT3 phosphorylation to show an enrichment in tumor-associated astrocytes of specimens of de-novo glioblastoma by immunostaining (Fig. 1e–g), as well as western blot of three patients with paired non-infiltration cortex and peritumoral region specimens and FACS analysis Supplementary Fig. 5.

**CD274+/GFAP+ astrocytes are enriched at the peritumoral glial scar.** We then performed immunohistochemical labelling on specimens from 43 glioblastoma patients with de-novo and recurrent glioblastoma to validate the presence of CD274+ astrocytes in the tumor environment, as well as GFAP and marker genes of various myeloid cell types. We identified CD274+ astrocytes in almost all samples (42 of 43 patients, 97.6%), with an

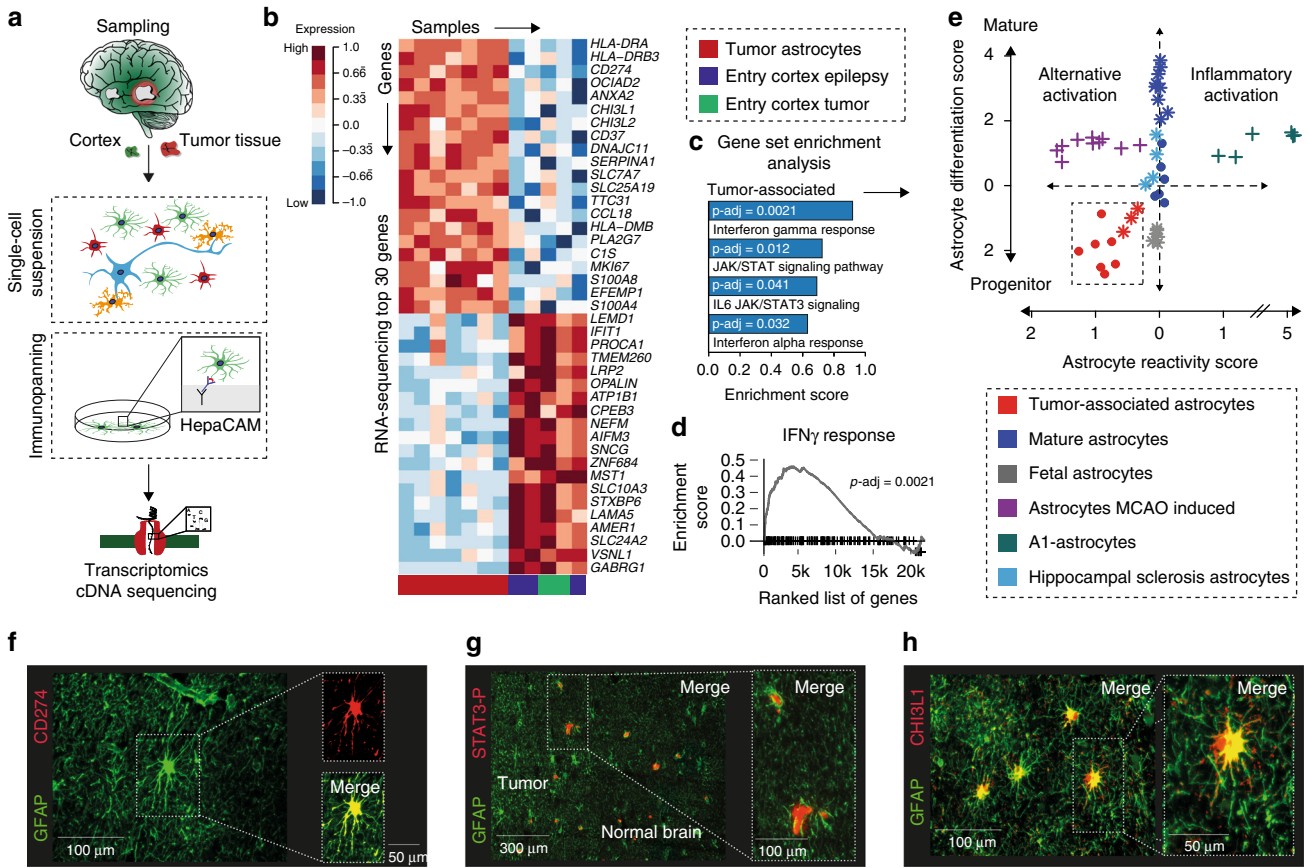

**Fig. 1** Purification and transcriptional profiling of tumor-associates astrocytes. **a** Illustration of the workflow. Cortex specimens from epilepsy patients ($n = 3$), entry cortex from glioblastoma patients ($n = 2$) and glioblastoma specimens ($n = 7$) were collected, followed by purification of astrocytes using immunopanning. RNA was analyzed by cDNA sequencing. **b** AutoPipe unsupervised cluster and heatmap of 30 most representative genes of astrocytes derived from healthy cortex and tumor specimens. **c** Gene set enrichment analysis (GSEA) highlights an increased response to cytokines, and JAK/STAT signaling in tumor-associated astrocytes. Exact values are given in the source file. **d** Gene set enrichment plot of ranked gene expression indicate the enrichment of the IFNγ response in tumor associated astrocytes. **e** Two-dimensional scatterplot of astrocytic differentiation and reactivity reveals a shift in the tumor-associated astrocytes towards the progenitor phenotype and alternative reactivity. Each individual dot represents a transcriptome profile. Round dots mark the expressions analyses generated in this study, other data are marked with stars (Zamanian et al., 2012)[17], as well as crosses (Zhang et al., 2016)[13]. Colors indicate the source of astrocytes as illustrated below. List of selected genes is given in the source file. **f–h** Immunostaining of tumor-associated astrocytes localized at the glial scar in the infiltrating region. GFAP+-astrocytes were marked by CD276 expression, STAT3 phosphorylation and CHI3L1 expression

exclusively increased number of CD274+/GFAP+ positive cells in the peritumoral glial scare (Fig. 2a, b). We further mapped the distribution of microglia (IBA1+, P2RY12+, and HLA-DR+), as well as macrophages/microglia (CD68+) and CD3+ cells in all regions. In comparison with reactive astrocytes, myeloid cells were not uniquely enriched in the peritumoral cortex, Fig. 2c, d.

**Astrocytic activation is driven by tumor and microglia cells**. Recent studies have shown that the inflammatory astrocyte subtype was mediated through microglial signaling[16,19], as astrocytes do not respond to inflammatory stimuli such as lipopolysaccharides (LPS)[16]. Thus, astrocytic activation is primarily caused by the crosstalk of the microglia and astrocytes. Whether this also applies to tumors is currently unexplored. In order to investigate to what extent tumor cells directly activate astrocytes and whether microglia is indeed involved in astrocytic activation, we have established an organotypic human culture model that closely mimics physiological human brain conditions. Murine models are often used to model human disease states and investigate the mechanisms that cause disease. However, the direct transfer of murine experimental data to human pathological events often fails[20] due to numerous

differences in the architecture of the immune system[21], astrocytes[13] and other cell types of both species. In order to replicate human environment, we did not make use of a murine model. Human slices were generated from specimens obtained either from entry cortex of epilepsy patients or non-infiltration entry cortex of glioma patients (Fig. 3a). Post sectioning, the slices were grown in serum-free growth medium. For microglia depletion 1 mg/ml of Clo-dronate[22] was added to the slice culture, detailed information of the time course is given in Supplementary Fig. 6. Maximal depletion was observed after 72 h of Clodronate administration (Fig. 3b). Once the depletion was complete, tumor cells labeled with ZsGreen from patient-derived stem-cell cultures were injected (20,000 cells) and cultured for 6 days (Fig. 3c). To assess the astrocytic activation due to tumor or microglia we purified astrocytes from the slices (Fig. 3a) and compared differentially expressed genes of control condition versus tumor injected slices in non-depleted (Microglia (+)) and microglia depleted slices (Microglia(−)), Fig. 3d. We identified an increased expression of genes that were previously found in the astrocytes from tumor specimens (Fig. 1b), (HLA-DRA, CHI3L1 SERPINA1, CCL18, CD37, and CD274). As a result of microglia depletion, we observed a less pronounced expression shift of astrocytes (as described above, Fig. 1) in reaction to tumor

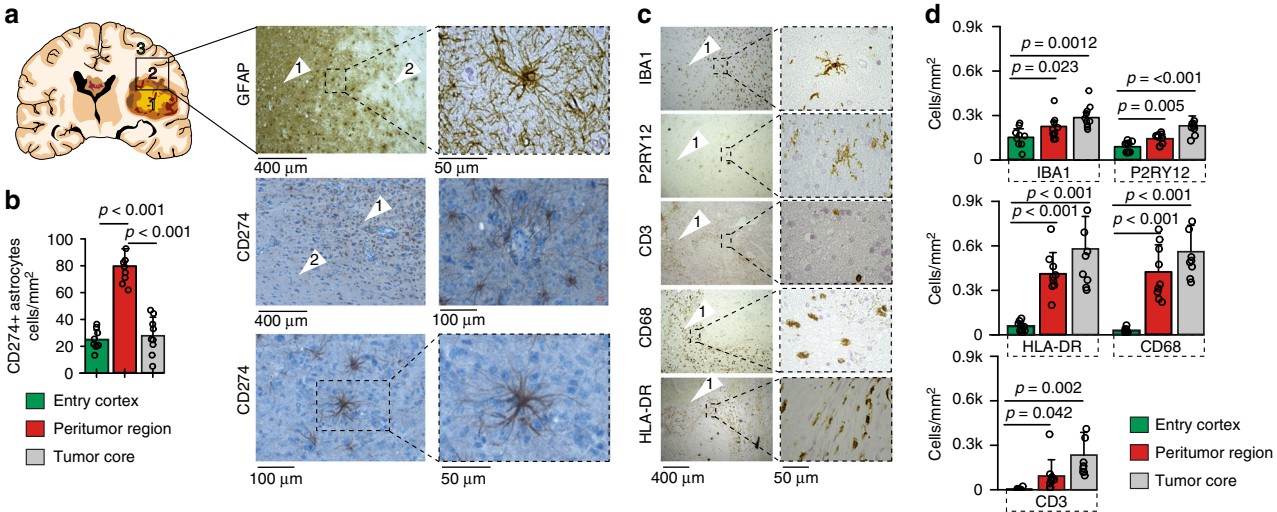

**Fig. 2** CD274+-astrocytes in glioblastoma specimens. **a** Immunohistochemistry of GFAP and CD274 of the tumor margin, arrows indicate the regions illustrated in the left panel. **b** Each dot represents the average number (3 fields per sample) of astrocytes per cm$^2$ in entry cortex ($n = 9$), peritumoral region ($n = 9$) and the tumor core ($n = 10$). Exact values, as well as statistical analysis are given in the source file. **c** Immunohistochemistry of the myeloid landscape of the tumor margin, microglia were marked by IBA1 and P2RY12, T-cells by CD3, tumor-associated macrophages/activated microglia by CD68 and HLA-DR. **d** Each dot represents the average number of cells per cm$^2$ in all different regions (3 fields per sample). Exact values, as well as statistical analysis are given in the source file. Arrow 1: Tumor Core; Arrow 2: Peritumoral Tissue and Infiltrating Region; Arrow 3: Non-Infiltrating Region. *P*-values are determined by one-way ANOVA (**d**) adjusted by Benjamini-Hochberger for multiple testing. Data is given as mean ± standard deviation

injection. Particularly, the strong JAK/STAT pathway activation was lost in microglia depleted slices, Fig. 3e. We further identified a loss of the IFNγ and IFNα pathway activation (Supplementary Fig. 7a, b), as well as a significant downregulation of genes regulated by IFNγ and IFNα (*IFI44L, ISG15, IFIT3, IFI6*), Supplementary Fig. 7c.

We then analyzed pro-inflammatory and anti-inflammatory cytokines from the medium (at day 6 after tumor injection) using an enzyme-linked immunosorbent assay, which revealed that an increased level of the anti-inflammatory IL10 was exclusively observed in slices containing microglia cells ($p < 0.001$) Fig. 3f. TGFβ, IFNγ, and G-CSF were found to be increased in tumor-injected slices without association with microglia depletion. Further, we aimed to validate STAT3-P and KI67 as a marker for reactive adaptation of tumor associated astrocytes by immunostainings and FACS analysis. In line with the expression data, we found an increased number of STAT3-P+ cells only at the concurrent presence of microglia cells, Fig. 3g, h. KI67+ cells were also observed in the presence of tumor in microglia depleted samples, suggesting that increased proliferation of reactive astrocytes is partially mediated by tumor cells. We used FACS analysis to analyze the heterogeneity of reactive astrocytes (STAT3-P+/ KI67+) within our slice model. We removed tumor cell contamination by gating of ZsGreen+ cells (Fig. 3j and Supplementary Fig. 8) and analyzed HepaCAM+/ZsGreen− cells. The results were then analyzed using t-distributed stochastic neighbor embedding statistics (t-SNE), resulting in four clusters (Fig. 3i). The first cluster contained non-tumor injected samples with and without microglia depletion showing low levels of STAT3-P and KI67 intensity. The tumor injected slices with depleted microglia were found to be divided into two separated clusters. One of these clusters revealed a slightly increased KI67 intensity but low levels of STAT3-P. The fourth cluster contained cells from tumor injected samples without microglia depletion, marked by highly increased STAT3-P and KI67 protein-level, Fig. 3k, l.

**Cytokine environment promote alternative reactive state.** Next, we aimed to investigate to what extent astrocytic activation is

caused by environmental factors. Therefore, we analyzed the RNA-seq profiles from an astrocyte cell line co-cultured beneath the slices without direct contact to the slices. Expression profiles were integrated into our reactive classifier model, which revealed a loss of expression shift towards the progenitor stage and maintained shift towards the *alternative activation*, Fig. 4.

**Microglia transcriptional profile mediated by activated astrocytes.** Our results suggest that microglia-astrocyte crosstalk plays a major role in the genesis of an immunosuppressive environment in glioblastoma, referred in the literature as "cold tumor"[23]. This raises the question to what extent astrocytes participate in the regulation of the immunosuppressive environment. To address this question, we aimed to further investigate the transcriptional adaptations of microglia from our slices with and without tumor injection. We purified microglia/myeloid cells (CD45+ cells) and performed expression analysis by RNAseq, demonstrating a typical microglia activation pattern, which was recently reported by others (for comparison we analyzed the data of Darmanis and colleagues[11] Supplementary Fig. 9), highlighted by increased expression of immune/APP's (acute phase proteins) related genes (*HLA-DRA, HLADRB1, HLA-DP1, HLA-LA, IFI30, APOE, APOC2*, Fig. 5a and Supplementary Fig. 10). Since astrocyte depletion is not feasible in our slice model due to high toxicity for multiple other cell types (neurons and microglia), we attempted to investigate the isolated influence of astrocytes on microglia/tumor crosstalk in a culture model, Fig. 5b. We did not detect any difference in the gene expression profiles between microglia cells cultured alone or in co-culture with astrocytes (Supplementary Fig. 11a–c). In line with our previous analysis, we compared differentially expressed genes in microglia/tumor co-culture with (Astrocytes (+)) and without (Astrocytes (−)) astrocytes, Fig. 5c. We observed that large parts of microglia activation are not affected by astrocytic co-culture. However, a subset of genes up-regulated in microglia/tumor co-culture was found to be exclusively related to the concurrent presence of astrocytes (*APOE, APOC2, HLA-DRA*). In addition, we identified an increased hypoxic metabolism marked by increased glycolysis

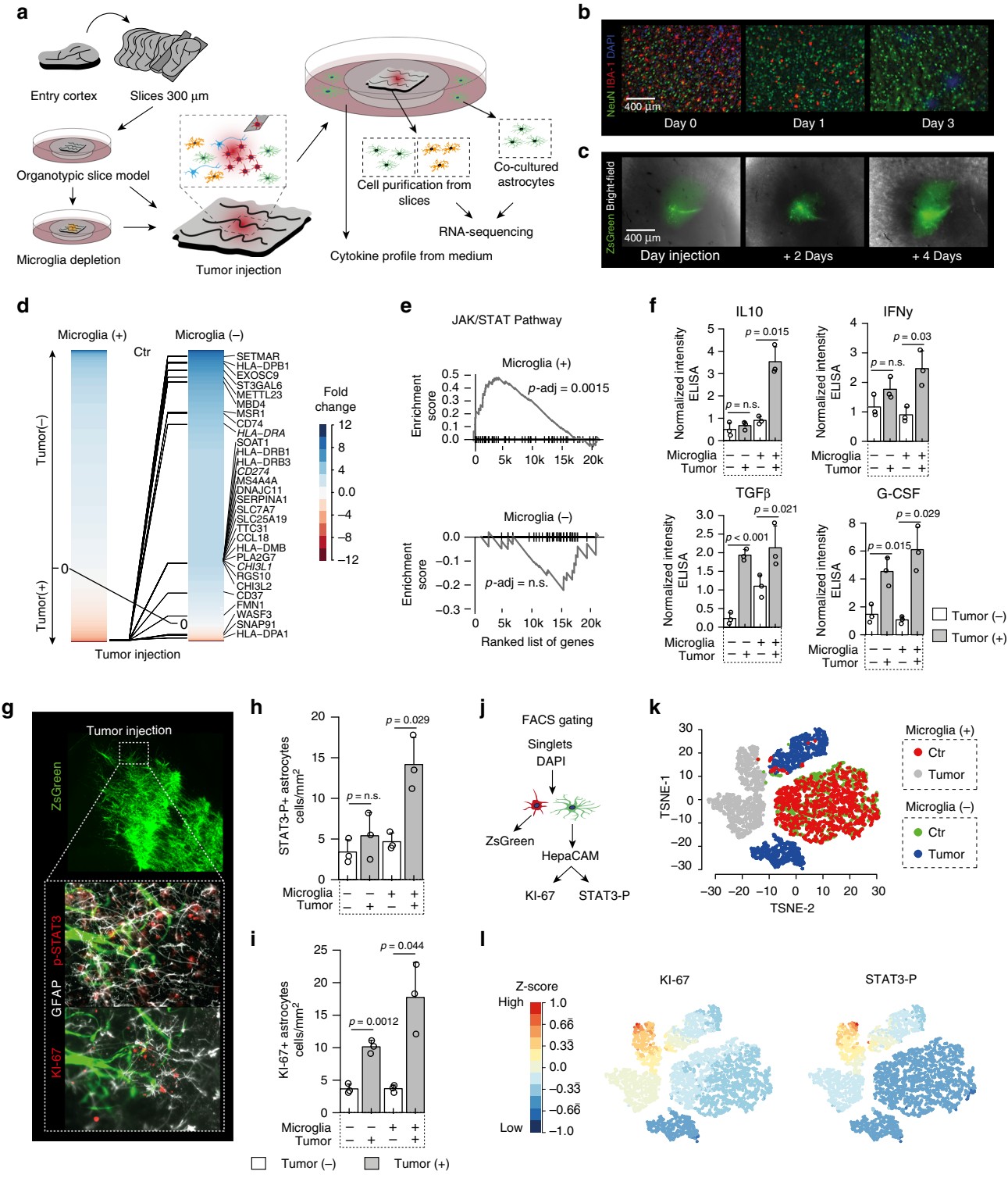

in microglia co-cultured with tumor cells, which is typically related to the inflammatory state of microglia[24]. This metabolic alteration was reduced by the additional presence of astrocytes, Fig. 5d. In line with our previous reported results, we were able to confirm the increased intensity of IL10 linked to microglia/astrocytes crosslink in the tumor environment, Fig. 5e–h.

**JAK-inhibition increase inflammation in glioma environment.** We hypothesized that the immunosuppressive environment is

mediated by astrocyte/microglia crosstalk and JAK/STAT activation of tumor-associated astrocytes lead to a high level of IL10, which maintains the anti-inflammatory environment. IL-10 is associated with a deficient anti-tumor immune response, increased TGF-β levels and feed-forward STAT3 signaling in tumor cells, microglia and probably astrocytes[24–26]. Inhibition of the centrally located JAK/STAT pathway potentially disperses the microenvironmental architecture followed by increased inflammation. We therefore analyzed the balance between pro-inflammatory and anti-inflammatory cytokines in slices that

**Fig. 3** Microglia loss-of-function model with transcriptional profiling of astrocytes. **a** Illustration of the workflow to set-up a human slice model combined with microglia depletion and tumor injection. Entry cortex was taken from the operation theatre, sliced within 10 min into 300 µm slices and cultured in serum-free conditions. In a 3 days time-course, slices are incubated with 1 mg/ml of Chlodronate to deplete microglia or control condition. 20,000 primary serum-free cultured ZsGreen tagged tumor cells were injected. After 4 days of tumor growth, astrocytes and microglia were purified for RNA-seq analysis (Detailed workflow is given in the Supplementary Fig. 6) **b** Representative staining of IBA-1 confirmed a robust depletion of microglia in human slices without loss of NeuN expression in neurons. **c** Representative staining of tumor injection in a time dependent manner. **d** Analysis of differentially expressed genes of purified astrocytes in Microglia(+) or Microglia(−) condition. Genes are ordered based on fold-change of gene expression between the control (Tumor(−)) and tumor injection (Tumor(+)), with blue indicating an >= 12 fold higher expressed in control samples and red an >= 12 fold higher expression in tumor injected samples. Lines indicate the differences of the fold-change rank (top 30 genes) between Microglia(+) and Microglia(−) condition. The R-code and detailed description is given in the source data. **e** Gene Set Expression Analysis (GSEA) of ranked gene expression of Microglia (+) or Microglia(−) condition indicate the enrichment differences of the JAK/STAT pathway. **f** Cytokine protein level in all conditions. Exact values, as well as statistical analysis are given in the source file. **g** Representative immunostaining of tumor infiltration after 4 days of culture and quantification **h**, **i**. Exact values of cell numbers are given in the source file. **j** Gating strategy to purify astrocytes from FACS data. Detailed plots for gating are given in the Supplementary Fig. 8. **k** FACS data analyzed by T-SNE, colors indicate the experimental conditions. **l** T-SNE map with STAT3-P (left) and Ki-67 (right) intensity, colors indicate the intensity (red: high intensity, blue: low intensity). *P*-values are determined by one-way ANOVA (**f**, **h**, **i**) adjusted by Benjamini–Hochberger (**f**, **h**, **i**) or False-Discovery Rate (**e**) for multiple testing. Data is given as mean ± standard deviation

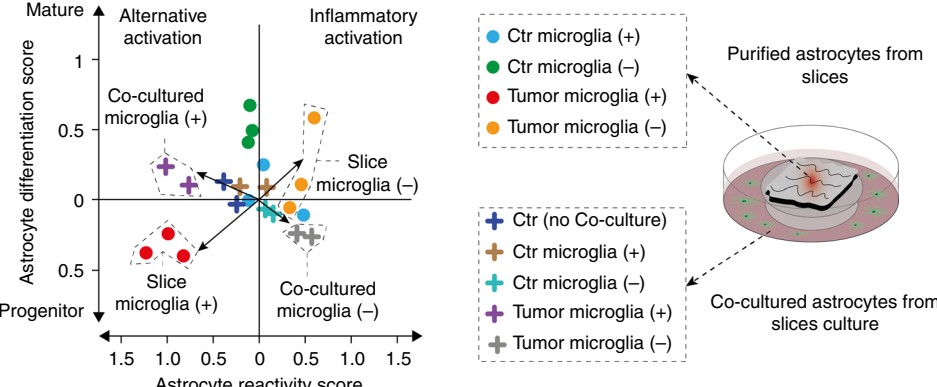

**Fig. 4** Transcriptional profiling of astrocytes from the human organotypic slice model. Scatterplot of astrocytic differentiation and reactivity reveals a shift in purified astrocytes from all experimental conditions and co-cultured astrocytes towards the progenitor phenotype and alternative reactivity. Astrocytes purified from slices are illustrated by dots, astrocytic cell lines marked by crosses. List of analyzed genes is given in the source file (Source File Fig. 1e)

were either pre-treated or post-treated with JAK inhibitor Ruxolitinib upon tumor injection, Fig. 6a. We found a significant reduction of tumor growth in pre-treated and post-treated JAK inhibited slices (Fig. 6a, b), as well as a reduced number of activated astrocytes marked by STAT3 phosphorylation within the tumor margin (Fig. 6d). Further, we found a significant up-regulation of several pro-inflammatory cytokines (TNFα, IL17A, IFNγ, IL4, IL2, IL6, IL12, IL13) with simultaneous loss of anti-inflammatory cytokines such as TGFβ and IL10, Fig. 6e, f. This effect was observed by concurrent tumor injection and JAK inhibition, as well as in pretreated slices, suggesting that the increased pro-inflammatory response is mainly caused by cells of the environment instead of tumor cells in first instance.

## Discussion

The dysregulation of inflammatory responses is involved in the pathogenesis and course of numerous diseases of the central nervous system (CNS), including traumatic, ischemic, degenerative, and malignant diseases[15]. In response to pathological alterations, astrocytes undergo a specific transformation called "astrogliosis" and adapt a "reactive state" which was shown to influence immune surveillance and inflammation[15]. In glioblastoma, a strong anti-inflammatory environment impairs a significant immune response which translates in a "cold" microenvironment and, therefore, into a low response to novel immune therapies[27]. Based on the fact that astrocytes potentially interact with immune cells and take part in the evolution of the anti-inflammatory environment, we aimed to specifically analyze

tumor-associated astrocytes and their reactive state. Through recent technical advances, it has been shown that astrocytes can undergo different reactive transformations based on age, spatial adaptation, and disease origin. Most recently, Zamanian and colleagues subclassified the reactive stages into an inflammatory subtype termed as "A1-specific" and a "helpful" subtype, which was described in a murine stroke model and named "A2-specific"[16,28,29]. These new categories were adopted from the activation states in microglia or macrophages (M1-, M2a/b/c), where the specific functions of these states are still an area of intense debate. In particular, the "A2-specific" phenotype of astrocytes is not clearly defined, and our results suggest that the "A2-specific" phenotype hides multiple different reactive stages, Fig. 1e. From a recent report, a small set of tumor-associated astrocytes was characterized by RNA-seq, which revealed similarities to fetal astrocytes[13]. In our study, we confirmed this transcriptional shift towards the progenitor stage, marked by increased proliferation and JAK/STAT pathway activation. We validated our results by comparison with scRNAseq data from Darmanis and colleagues[11], Supplementary Fig. 4. In order to selectively purify astrocytes from tumor specimens, we used an immunoprecipitation method called "immunopanning", recently described by Zhang and colleagues. The surface marker Hepatic And Glial Cell Adhesion Molecule (HepaCAM) has been described in numerous studies as being exclusively expressed in astrocytes[11–13,30]. Although Zhang and colleagues[13] used HepaCAM-based immunopanning to purify astrocytes from tumor specimens, currently the specificity of HepaCAM to purify tumor-associated astrocyte

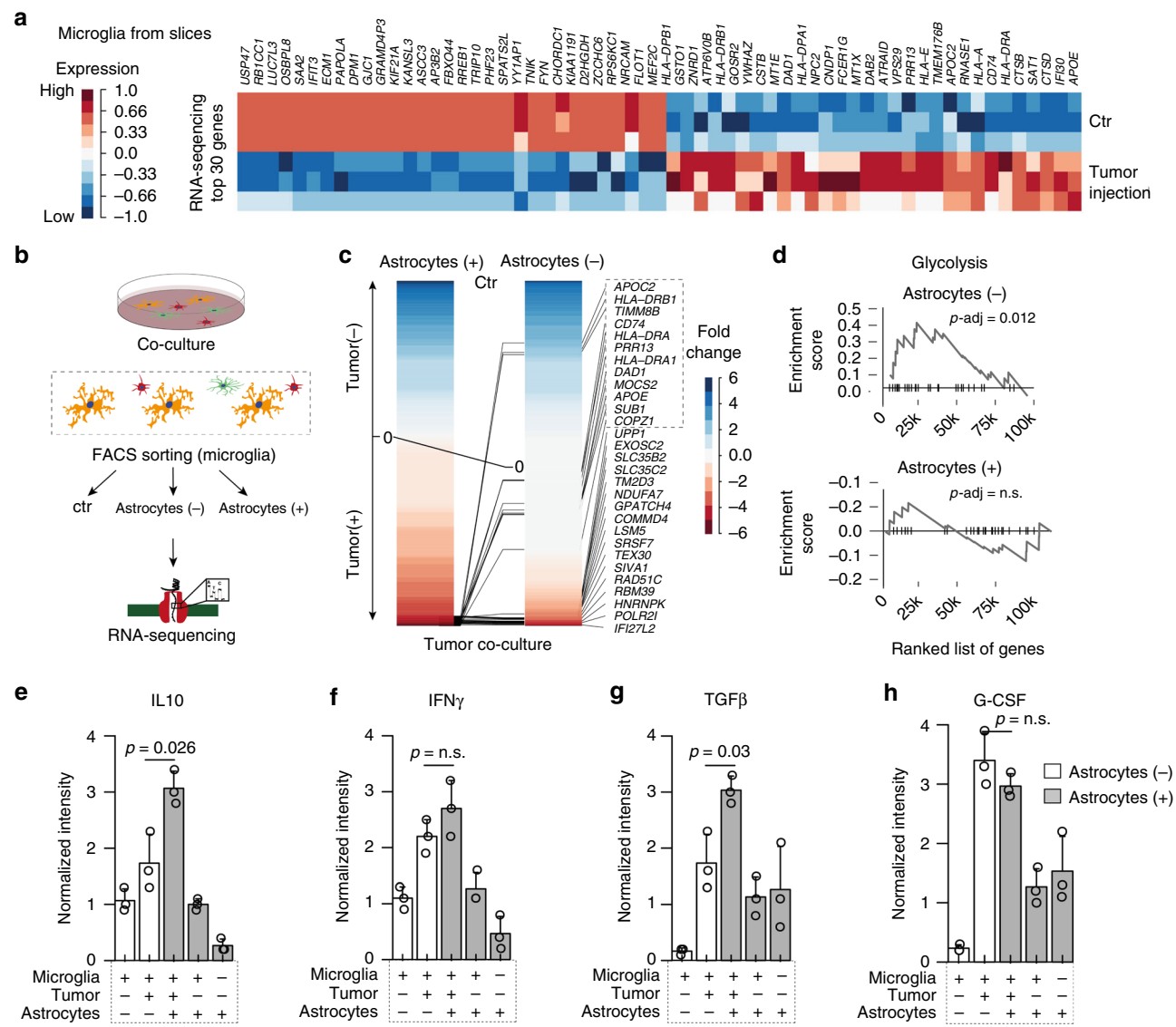

**Fig. 5** Transcriptional profiling of microglia from slices and a co-culture model. **a** AutoPipe unsupervised cluster and heatmap of 30 most representative genes expressed in microglia derived from control and tumor injected slices. **b** Illustration of the co-culture model. **c** Analysis of differentially expressed genes of purified microglia in Astrocytes (+) or Astrocytes (−) condition. Genes are ordered based of fold-change of gene expression, colored in blue for highly expressed in control samples and red for increased expression in tumor injected samples. Lines indicate the differences of the fold-change rank (top 30 genes) between Astrocytes (+) and Astrocytes (−) condition. **d** Gene Set Expression Analysis (GSEA) of ranked gene expression of Astrocytes (+) or Astrocytes (−) condition indicate the enrichment differences of the genes involved in glycolysis. **e–h** Bar plots of cytokine protein level in all conditions. Exact values, as well as statistical analysis are given in the source file. *P*-values are determined by one-way ANOVA **e–h** adjusted by Benjamini–Hochberger (**e–h**) or False-Discovery Rate (**d**) for multiple testing. Data is given as mean ± standard deviation

and the extent of tumor cell contamination remain unclear. For this purpose, we called copy number variations of purified astrocytes and tumor cells in order to prove low rates of tumor cell contamination, Supplementary Fig. 1c. Purified astrocytes from tumor specimens revealed a reactive state marked by IL10 and IFNγ response resulting in a JAK/STAT pathway activation. It has been described that enforced enhancement of JAK/STAT signaling leads to early astrogliogenesis and promotes differentiation of astrocytes[31]. In the course of traumatic injuries, STAT3-activated astrocytes were found to express axon growth-promoting molecules contrary to prevailing dogma that glial scars avoid axon regeneration[32]. These findings suggest that STAT3-activated astrocytes support the growth and regeneration ability of neurons. Further, the investigation of Priego and colleagues highlights the particular role of STAT3 labeled reactive astrocytes

in the tumor environment of brain metastasis. These subtype of reactive astrocytes significantly contributes to maintain the anti-inflammatory environment and promote tumor growth and invasion[18]. Assuming that the JAK/STAT activation of astrocytes has a direct effect on the immunosuppressive environment, an inhibition of this pathway should lead to a transformation from an anti-inflammatory to a pro-inflammatory environment. In line with our hypothesis, it was recently shown that the blockade of IFNγ signaling in astrocytes alone leads to a reduced inflammatory reaction, whereas the blockage of IFNγ signaling in microglia supports inflammation in an autoimmune encephalitis model[33]. Our work confirms the hypothesis that the anti-inflammatory environment results from a complex interaction between microglia and astrocytes. We further identified CD274 (PDL1, Programmed Death ligand 1) expressed by tumor-associated

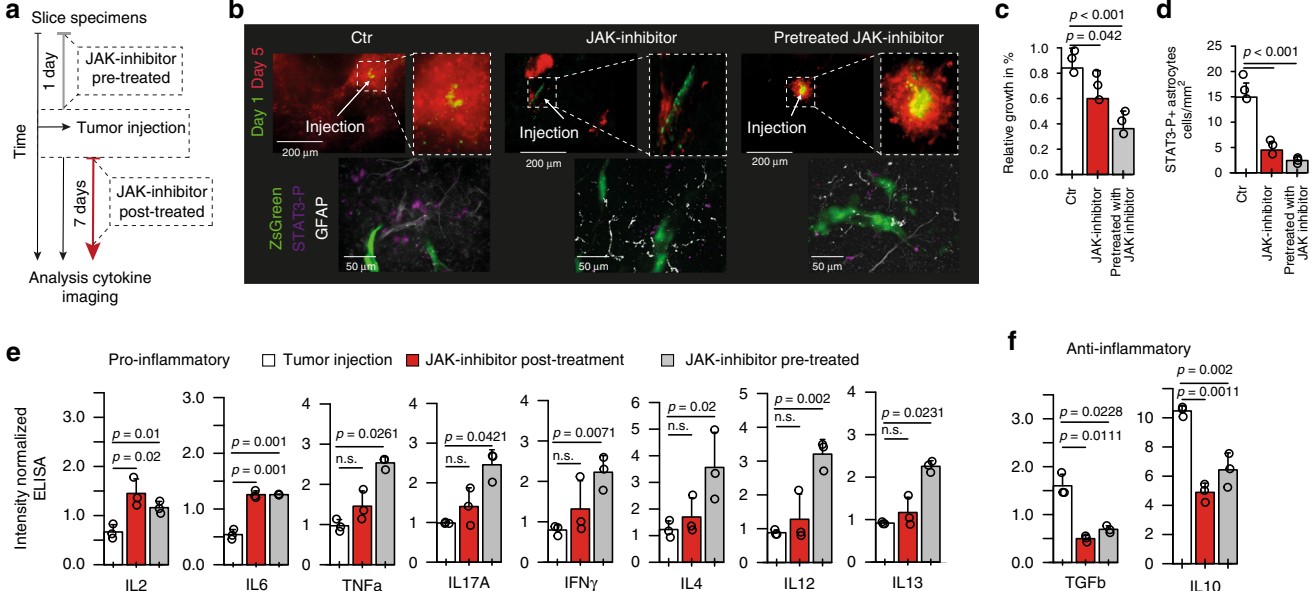

**Fig. 6** Cytokine profiling after JAK-inhibition. **a** Illustration of the workflow **b** Fluorescence imaging of the tumor growth in Ctr, JAK inhibition and pre-treated JAK-inhibitor slices condition. The time point of tumor injection is colored in green, the 5th day time point in red. On the bottom, immunostainings of the STAT3-P level in all experimental conditions. **c** Tumor growth in slices in all experimental conditions. **d** Number of STAT3-P cells per cm$^2$. **e** Level of pro-inflammatory and anti-inflammatory cytokines in tumor injected slices administrated to either control conditions or JAK-Inhibitor (Ruxolitinib). **f** Level TGFb and IL10 in different treatment regimes. $P$-values were determined by one-way ANOVA adjusted by Benjamini–Hochberger for multiple testing. Data is given as mean ± standard deviation. Exact values of all cytokines are given in the source file

astrocytes, which is a target of the JAK/STAT pathway and as an immune checkpoint provides immune suppression through the PD1-PDL1 axis[34]. This CD274 expression in astrocytes has also been described in inflammatory diseases such as multiple sclerosis[35]. Gliomas have long been considered inaccessible to anti-tumor immune response as they are largely protected from infiltrating immune cells. In addition, the microenvironment of gliomas, with their anti-inflammatory character, represents a barrier that has not yet been overcome. So far, the promising immunological therapies of other cancer diseases turned out ineffective for glioma treatment[36,37]. In this context, GBM are also referred to as immunologically "cold" tumors and the conversion into "warm" tumors would be an important step towards the establishment of new immunological therapies. The most important cytokines that suppress the immune response are tumor growth factor beta (TGFβ), interleukin 10 (IL10) and colony stimulating factor (CSF)[36]. Here, we aimed to investigate the role of the individual cellular components that are involved in the evolution of the anti-inflammatory environment. Our organotypic model attempts to closely simulate the environment of the human brain to mimic physiological conditions. We were able to deplete microglia and study the isolated role of human astrocytes. We showed that the secretion of anti-inflammatory TGFβ is not affected by microglia cells, although IL10 was primarily released by the coexistence of both, microglia and astrocytes, suggesting that astrocytes mediate a specific re-programming of microglia cells. Further, we demonstrated that distinct tumor-associated microglia activation was maintained by the crosstalk of tumor, astrocytes and microglia crosstalk. These findings were limited by the necessary use of immortalized cell lines, but concurred with the results obtained from the human slice model above. Our findings suggest that astrocytes support the secretion of anti-inflammatory cytokines which are known to inhibit T-cell activation and prevent an efficient immune response[36]. By pharmacological targeting of the JAK/STAT pathway by Ruxolitinib, an FDA-approved JAK inhibitor, we were able to

overcome the immunosuppressive environment towards increased pro-inflammatory environment. In addition, we showed that tumor cell spread was significantly reduced especially if slices were pretreated by JAK inhibitor. The reduced number of reactive astrocytes by JAK inhibitor[38], as well as reduction of tumor growth and increased recruitment of myeloid cell within the tumor was recently investigated[39]. In summary, we reported the immunomodulatory properties of tumor-associated astrocytes within the human glioblastoma environment. The reactive state of tumor-associated astrocytes is driven by the JAK/STAT3 pathway and increased proliferation. One novel aspect this work is the recognition that the evolution and maintenance of "cold" environment requires crosstalk between various cell entities, in particular reactive astrocytes. This interaction results in an increased release of the anti-inflammatory cytokines TGFβ, IL10, and CSF. The astrocyte-microglia interaction within the tumor environment mediates transcriptional re-programming in microglia and myeloid cells towards an anti-inflammatory state. Finally, these effects promote an overall anti-inflammatory environment, potentially hindering immune therapeutic approaches. In addition to immune checkpoint inhibitors, targeting specific environmental components may improve the efficiency of immune therapies.

## Methods

**Ethical approval and resource sharing**. The local ethics committee of the University of Freiburg approved data evaluation, imaging procedures, and experimental design (protocol 100020/09 and 472/15_160880). The methods were carried out in accordance with the approved guidelines, with written informed consent obtained from all subjects. The studies were approved by an institutional review board. Further information and requests for resources, raw data, and reagents should be directed and will be fulfilled by the Contact: D. H. Heiland, dieter.henrik. heiland@uniklinik-freiburg.de. Full table of all materials is given in the supplementary information.

**Human organotypic slice culture**. Reagent sources for all cell culture and flow cytometry experiments are listed in Table 1. Capillaries and damaged tissue were

## Table 1 Reagent sources

| Reagents/resources | Source | Identifier |
|---|---|---|
| **Antibodies** | | |
| AffiniPure Goat Anti-Mouse IgG + IgM | Jackson ImmunoResearch Laboratories Inc., West Grove, PA, USA | 115–005–044 |
| AffiniPure Goat Anti-Rabbit IgG (H + L) | Jackson ImmunoResearch Laboratories Inc., West Grove, PA, USA | 111–005–003 |
| Anti-CD11b (Rabbit) | Abcam, Cambridge, UK | ab52478 |
| Anti-CD45 (Rabbit) | Abcam, Cambridge, UK | ab10558 |
| Anti-CD68 (Mouse) | Abcam, Cambridge, UK | ab201340 |
| Anti-HepaCAM (Human) | R&D Systems, Minneapolis, USA | MAB4108 |
| Anti-STAT3 (Rabbit) | Abcam, Cambridge, UK | ab30647 |
| Anti-STAT3-P (Rabbit) | Abcam, Cambridge, UK | ab76315 |
| Anti-GFAP (Rabbit, donkey) | Dako, Santa Clara, USA; Sigma, St. Louis, Missouri, USA | Z0334, G9269 |
| Anti-TGFB (Rabbit) | Abcam, Cambridge, UK | ab92486 |
| Anti- NeuN (Mouse) | Millipore, Massachusetts, USA | MAB377 |
| Anti- IBA-1 (Rabbit) | Wako, Richmond, USA | 019-19741 |
| Anti-CD11b (Rabbit) | Abcam, Cambridge, UK | Ab133357 |
| Anti-α-Tubulin (Mouse) | Santa Cruz Biotechnology, Texas, USA | Sc-8035 |
| Anti- Ki67 (Rabbit) | Abcam, Cambridge, UK | Ab15580 |
| DAPI | Sigma, Missouri, USA | 32670 |
| Goat anti-Mouse IgG Alexa Fluor 488 | Life Technologies Coorperation Eugene, USA | A11001 |
| Goat anti-Rabbit IgG Alexa Fluor 568 | Life Technologies Coorperation Eugene, USA | A11011 |
| Donkey anti-Goat IgG Alexa Fluor 647 | Life Technologies Coorperation Eugene, USA | A21447 |
| Goat anti Rabbit IgG Alexa Fluor 488 | Life Technologies Coorperation Eugene, USA | A11008 |
| Donkey anti-rabbit IgG Alexa Fluor 555 | ThermoFisher Scientific, Massachusetts, USA | |
| Goat anti-Mouse IgG-HRP | Santa Cruz Biotechnology, Texas, USA | Sc-2005 |
| Goat-anti-Rabbit IgG-HRP | Santa Cruz Biotechnology, Texas, USA | Sc-2004 |
| Mouse-anti-Goat IgG-HRP | Santa Cruz Biotechnology, Texas, USA | Sc-2354 |
| PE/CY5 conjugation kit | Abcam, Cambridge, UK | Ab 102893 |
| APC/CY7 conjugation kit | Abcam, Cambridge, UK | Ab 102859 |
| Ki-67 efluor 660 (SolA15) | ThermoFisher Scientific, Massachusetts, USA | 50-5698-82 |
| **Chemicals** | | |
| Neurobasal™ medium(L-Glutamine) | Gibco, Massachusetts, USA | 21103049 |
| Hibernate-A™ medium | Gibco, Massachusetts, USA | A1247501 |
| D+Glucose | Sigma, Missouri, USA | G8644 |
| N-methyl D-Glucamine (NMDG) | Sigma, Missouri, USA | M2004 |
| Glutamax™ supplement | Gibco, Massachusetts, USA | 35050061 |
| B-27™ Supplement (50×) | Gibco, Massachusetts, USA | 17504001 |
| Antibiotic-Antimycotic (Anti-Anti) (100×) | Gibco, Massachusetts, USA | 15240062 |
| Magnesium sulfate (MgSO₄) | Sigma, Missouri, USA | M3409 |
| HEPES | Sigma, Missouri, USA | H0887 |
| TRIzol™ reagent | ThermoFisher Scientific, Massachusetts, USA | 15596026 |
| Clodronate disodium (Dichloromethylenediphosphoric acid disodium salt) | Sigma, Missouri, USA | D4434 |
| Ruxolitinib | Adipogen | AG-CRI-3624 |
| **Insert** | | |
| Millicell inserts (30 mm, 0.4 μm) | Millipore | PICMORG50 |

dissected away from the tissue block in the preparation medium containing: Hibernate medium supplemented with 13 mM D + Glucose, 30 mM NMDG and 1 mM Glutamax. Three hundred micrometer thick coronal slices sectioned using a vibratome (VT1200, Leica Germany) were incubated in preparation medium for 10 min before plating to avoid any variability due to tissue trauma. Up to three to four slices were gathered per insert. The transfer of the slices was facilitated by a polished wide mouth glass pipette. Slice were maintained in growth medium containing Neurobasal (L- Glutamine) supplemented with 2% serum free B-27, 2% Anti- Anti, 13 mM D+ Glucose, 1 mM $MgSO_4$, 15 mM HEPES (Sigma, H0887), and 2 mM Glutamax at 5% $CO_2$ and 37 °C. The entire medium was replaced with fresh culture medium 24 h post plating and every 48 h thereafter. Medium was always collected and frozen at −20 °C for ELISA cytotoxicity measurement.

### Chemical depletion of microglia from slice cultures.
Selective depletion of microglia in organotypic brain slices was achieved by supplementing the growth medium with 1 mg/ml Clodronate (Sigma, D4434) for 72 h at 37 °C. Subsequently, the slice cultures were carefully rinsed with growth medium to wash away any debris. Before and end of microglia depletion medium was always collected and frozen at −20 °C for ELISA measurements.

### JAK inhibition in organotypic brain slices.
JAK inhibition is organotypic brain slices was achieved by supplementing the growth medium with 30 μM Ruxolitinib for either 72 h or in the presence of inhibitor at 37 °C. Finally, the slices were used for further experimental conditions and the medium was always collected and frozen at −20 °C for ELISA measurements.

### Slice/astrocyte co-culture model.
Inserts with the slices (either with or without microglia) were transferred to the plate containing 300,000 astrocyte cells growing in a serum-free growth medium (as mentioned above). At the end of the experiment, the medium was carefully aspirated and frozen at −20 °C and cells were briefly scraped by adding 300 μl of TRIzol to each well. Then, the TRIzol/cell lysate was carefully transferred to 1.5 ml eppendorf tubes and kept at −20 °C until RNA preparations.

### Tumor injection onto tissue cultures.
*ZsGreen* tagged GBM cell lines cultured and prepared as described in the cell culture section. Post trypsinization, a centrifugation step was performed, following which the cells were harvested and suspended in MEM media at 20,000 cells/μl. Cells were used immediately for injection onto tissue slices. A 10 μL Hamilton syringe was used to manually inject 1 μL into the white matter portion of the slice culture. Slices with injected cells were incubated at 37 °C, 5% CO2 for 7 days and fresh culture medium was added every 2 days. Medium was collected and frozen at −20 °C for ELISA measurement.

Tumor proliferation was monitored by imaging at day 0, 4, and 7 by using an inverted fluorescence microscope (Zeiss, Observer D.1). After 7 days of culture, slices were fixed and used for immunohistochemistry.

### Cell culture and co-culture model.
Astrocytes (CRL-8621), Tumor cell lines (Glioma Stem-like Cells, GSC 1, 2, and 3, purified from surgical specimens) and microglia cells (T0251) were cultured in serum-free conditions as described previously[40]. For combining different cell types within a co-culture, we used fluorescence tagged cells and culture them under serum-free condition along a time period of 48 h. Cells were plated on laminin-coated dishes as described previously[40]. Validation of co-culture was done after 24 h and 48 h by fluorescence microscopy.

### Cell purification by immunopanning.
Immunopanning was performed according to a modified version of the purification protocol from Zhang et al.[13]. The generation of a single-cell suspension from tumor tissue for subsequent immunopanning was achieved by using the Worthington Papain Dissociation System (Worthington Biochemical Corporation, Lakewood, NJ, USA) according to the manufacturer's instructions. Before each use, the reagents of the kit were equilibrated with a 95%O2:5%CO2 gas mixture to preserve a constant pH value. The dissociation steps were performed under sterile conditions in a cell culture bench. The resected tissue was placed in a culture dish filled with PBS and was reduced to a total volume of 0.5 cm$^3$ before fragmentation to smaller pieces. The supplied papain vial was then reconstituted with 5 ml EBSS to obtain a solution with 20 units papain per ml in 1 mM L-cysteine with 0.5 mM EDTA. Brief incubation in a 37 °C water bath was required for optimal solubility and enzymatic activity. The provided DNase vial was resuspended in 500 μl EBSS to yield a solution with 2000 units of deoxyribonuclease per ml. Two hundred and fifty microliter of the resulting solution was added to the prepared papain mix to obtain a final concentration of 0.005% DNase. The tissue fragments were then transferred to the papain solution, treated with 95%O2:5%CO2 and incubated for 90 min at constant agitation in a 37 °C water bath. Afterwards, the mixture was carefully triturated with a 10 ml pipette to obtain a high cell yield. The resulting cell suspension was separated from undissociated tissue fragments and centrifuged at $300 \times g$ for 5 min. Before the first use, the supplied albumin-ovomucoid protease inhibitor was reconstituted with 32 ml of EBSS to obtain a final concentration of 10 mg inhibitor and 10 mg albumin per ml. After centrifuging, the cell pellet was resuspended in a solution with 2.7 ml EBSS, 300 μl inhibitor and 150 μl of the previously prepared DNase solution. The isolation of intact cells was achieved by pipetting the cell suspension on top of 5 ml albumin-ovomucoid inhibitor and thus preparing a discontinuous density gradient. After centrifuging at 70 g for 6 min, the resulting cell pellet was used for immunopanning. For the set up of panning plates, 15 cm plastic petri dishes were coated with 25 ml of 50 mM Tris-HCL pH 9.5 and 60 μl of the secondary antibody (anti-mouse IgG+IgM and anti-rabbit IgG, respectively). The panning plates were incubated at 4 °C overnight and washed three times with PBS before adding the cell type specific primary antibody (20 μl of anti-CD45 and 15 μl of anti-HepaCAM in 12 ml 0.2% BSA, respectively). The panning dishes were incubated at room temperature for at least 2 h or at 4 °C overnight. The coated plates were washed three times with 0.2% BSA before use. Tumor tissue samples were obtained during neurosurgical resections and dissociated with papain as described above. The resulting cell pellet was directly resuspended in panning buffer and filtered through a 100 μm cell strainer in order to ensure a homogeneous single-cell suspension without cell clumps. The cell suspension was then added to the CD45 panning plate and incubated at room temperature for 30 min. The incubation time was interrupted after 15 min for a short period

of constant agitation. The unbound cells were then transferred to the subsequent HepaCAM plate for incubation. The CD45 plate with bound microglia/macrophages and the HepaCAM plate with bound astrocytes were washed eight times with PBS immediately after incubation to ensure highly specific bindings. The successful capture of cells was visually examined under a microscope. The bound cells were then scraped directly off the panning dish with 1 ml of Qiazol reagent (Qiagen).

**Viral transduction by constitutive reporter lentiviral vectors**. For whole-cell tracking, tumor and astrocytes, primary cultured glioblastoma cells and Astrocytes were transduced with lentiviral particles (astrocytes: *pmCherry* Vectror, Clonetech, GSC: *pZsGreen1-1* Vector, Clonetech). For the transduction $3 \times 10^6$ cells were seed per well and incubate overnight in a 37 °C, 5% $CO_2$ incubator. Particles quantity determination was calculated according to the manufacturer's instructions. The transduction mix was prepared by adding the required volume of thawed viral particles and Polybrene® (800 μg/ml). Medium was changed after 1 day. Quality of transduction was measured after 2 days.

**Flow cytometry**. For cells: Astrocytes (mCherry) and tumor cells (ZsGreen) or microglia (AmCyan) were sorted by FACS Aria III (BD Bioscience) in the core facility, Medical-Center Freiburg, University of Freiburg in accordance to the manufacturer's instructions.

For slices: Tissue specimens were mechanically dissociated using a glass homogenizer on ice and sequentially passed through 100 μm and 40 μm nylon cell strainers (BD Falcon #352360 and #352340). The mesh was then rinsed several times with 4 °C cold PBS/EDTA. Resulting cell suspensions can be kept on ice for up to 20 min while other tissue samples are being processed. After centrifugation ($310 \times g$; 4 °C; 6 min) and removal of the supernatant, the cell pellet was suspended in 0.5 ml 4 °C cold PBS/EDTA. Five ml of −20 °C cold 80% methanol was added drop-wise under constant, gentle vortexing. Samples were incubated for 30 min on ice and subsequently overnight at −20 °C before being subjected to staining. Alternatively, samples can be stored at −20 °C for upto 1 year. The cell suspension was washed and centrifuged at $350 \times g$ for 5 mins and cells were counted to be roughly $2 \times 10^6$. This is further followed by resuspending the cells using permeabilization buffer (0.1% Triton X-100 in 1× PBS) for 5 mins at room temperature. Samples were centrifuged briefly and the pellets were washed 2 times with 1× PBS. Five microliter of TruStain FcX™ were added per million cells in 100 μl staining volume to avoid unspecific antibody binding (It is not necessary to wash the cells between these blocking and immunostaining steps. Cells were stained with fluorochrome-conjugated antibodies). Antibodies were directly conjugated with the following fluorescent tags: PE/Cy5 and APC/Cy7. The following antibodies were used: Anti-STAT3, HepaCAM, Ki67 efluor 660 and DAPI. Antibody staining was performed according to the manufacture instructions. Finally, cells were washed and resuspended in at least 0.5 to 1 ml of FACS buffer depending on the number of cells. We used Sony SP6800 spectral analyzer and recorded 100,000 events per sample in standardization mode with PMT voltage set to maximum to reach the saturation rate below 0.1%. Gating was performed by FCS Express 6 plus at the core facility, University of Freiburg as shown in the Supplementary Fig. 8.

**Secretome profiling**. For cytokine analysis, we used the Proteome Profiler Human XL Cytokine Array Kit (R&D Systems, Minneapolis, MN, USA). Cells were cultivated for 48 h in either mono-cultured or co-cultured conditions. Medium was harvested and centrifuged to remove particulates. The supernatant was analyzed according to the manufacturer's protocol. The resulting membranes were developed using ChemiDOC XRS with exposure of 20 min. For the evaluation of the signal intensity, we measured the pixel density with additional software in Image J in accordance to the protein array analyzer macro[41] and post-processed in R.

**Immunoblotting**. Cells were lysed using Radio Immuno Precipitation Buffer (RIPA buffer) and protease inhibitor on ice. Laemmli buffer was added to the samples and the concentration was adjusted. For western blotting 4–20% precast gels from BioRad or self-made polyacrylamide gels were used. A digital imager ChemiDoc XRS detected the chemiluminescence emanation from the membrane by transforming the signal into a digital image.

**Immunofluorescence**. For immunostaining, cells were grown on slides and fixed with 3% formaldehyde for 10 min at room temperature. Afterwards the slides were incubated in permeabilization solution (HEPES, Sucrose, NaCl, MgCl₂, 0.5% Triton X-100) for 20 min at 4 °C and subsequently blocked in PBS 2% BSA for 30 min at 37 °C. The slides were then washed three times for five minutes with PBS. The primary antibody was diluted in blocking buffer (2% BSA, PBS) and incubated for 90 min at 37 °C. After 3 washes (5 min) in PBS the secondary antibodies were diluted in blocking buffer and added to the coverslips for 45 min at 37 °C. After another 3 wash cycles a counterstaining with DAPI for 20 min followed. After another wash in water the coverslips were fixed on a glass plate. A Fluoview FV10i confocal microscope from Olympus was used for fluorescence microscopy. All measurements and image processing were performed using the company's software. Optical magnification settings of ×10 and ×60 with oil were

used. Laser power was manually optimized and used with equal settings for all imaged samples.

**Immunostaining for slice cultures**. The same protocol was followed for slices with or without microglia and tumour injection. Media was removed and exchanged for 1 ml of 4% paraformaldehyde (PFA) for 1 h and further incubated in 20% methanol in PBS for 5 min. Slices were then permeabilized by incubating in PBS supplemented with 1 % Triton (TX-100) overnight at 4 °C and further blocked using 20% BSA for 4 h. The permeabilized and blocked slices were then incubated in the following primary antibodies: anti- NeuN rabbit for neuronal nuclei (1:1000), anti-GFAP rabbit for astrocytes (1:2000), Iba-1 (1:1000, rabbit), Anti-CD11B (1:1000) for microglia, anti-STAT3 for reactive astrocytes and Anti-Ki67 marker for proliferation in 5% BSA-PBS incubated overnight at 4 °C. After washing in PBS, slices were labelled with secondary antibodies conjugated with Alexa 405, 488, 555, 568 for 3 h at room temperature. Finally, slices were mounted on glass slides using DAPI fluoromount (Southern Biotech, Cat. No. 0100–20).

**RNA sequencing**. The purification of mRNA from total RNA samples was achieved using the Dynabeads mRNA Purification Kit (Thermo Fisher Scientific, Carlsbad, USA). The subsequent reverse transcription reaction was performed using SuperScript IV reverse transcriptase (Thermo Fisher Scientific, Carlsbad, USA). For preparation of RNA sequencing, the Low Input by PCR Barcoding Kit and the cDNA-PCR Sequencing Kit (Oxford Nanopore Technologies, Oxford, United Kingdom) were used as recommended by the manufacturer. RNA sequencing was performed using the MinION Sequencing Device, the SpotON Flow Cell and MinKNOW software (Oxford Nanopore Technologies, Oxford, United Kingdom) according to the manufacturer's instructions. Samples were sequenced for 48 h on two flow-cells acquire a total sequence length of 16.38 Gbp (mean read length 689 bp, ~4 million reads). Basecalling was performed by Albacore implemented in the nanopore software. Only $D^2$-Reads with a quality Score above 8 were used for further alignment.

**Sequence trimming and alignment**. In the framework of this study, we developed an automated pipeline for nanopore cDNA-seq data, which is available at github (https://github.com/heilandd/NanoPoreSeq). First, the pipeline setup a new class termed "Poreseq" by a distinct sample description file.

```
#Set up a object from "Poreseq" class
Set = Poreseq(Samples_discription)
```

The analysis starts by rearrange reads from *fastq* output from the nanopore sequencer containing all $D^2$-Reads. All *fastq* files need to be combined into one file.

```
# Combine fastq files
GET_FASQ(Set)
```

Now multiplexed samples need to be separated in accordance to their barcode and trimmed by Porechop (https://github.com /rrwick/Porechop).

```
# Multiplex Samples
TRIM_Barcodes(Set)
```

Alignment was performed by minimap2 (https://github.com/lh3/minimap2) and processed by sam-tools.

```
# Alignment der Sequences
Set = NANOPORE_Aligner(Set)
```

Mapped reads were normalized by DESeq[42]. The expression matrix was analysed with AutoPipe (https://github.com/heilandd/AutoPipe) by a supervised machine-learning algorithm and visualized in a heatmap.

```
#Analyze Data
Set = Analyzer(Set,Write_exp = T,filter_genes = 50,MA_PLOT = T)
```

**Cluster based on distinct gene-set expression**. We first extracted the top 50 genes differentially expressed in astrocytes in fetal, adult, A1-activated and A2-activated state. Based on the mean expression of this extracted genes, new samples will be classified across above descripted stages. We submitted the code to github (https://github.com/heilandd/Code-Request/blob/master/PLOT_4D%20Function) for a broad availability.

```
#Add data into class "Astrocytes"
object = Astrocytes(data = as.data.frame(dat_exp))
```

In a first step, new data will be integrated into the class "Astrocytes", further, gensets will be added and data could be analysed.

```
#Save list with top ranked genes from A1,A2,f,a genes
list_genes = readRDS("List_genes.R")
object@A1_genes = List_genes[[1]]
object@A2_genes = List_genes[[2]]
object@fetal_genes = List_genes[[3]]
object@adult_genes = List_genes[[4]]
PLOT_4D(object)
object = Compute_Scores(object)
```

**Differential gene expression analysis**. Analysis of differentially expressed genes or differential metabolic intensities was performed using the DESeq2 package. The algorithm mainly uses a generalized linear model with a negative binomial distribution. A detailed description is given in the R documentation.

**Gene expression analysis of multiple groups and clustering**. In order to analyze transcriptome data of multiple samples we calculate the optimal number of clusters based on the "Partitioning Around Medoids" algorithm. A t-distributed stochastic neighbor embedding (tSNE)[43] analysis and Consensus Cluster[44] is performed for validation. To identify the core samples of each cluster, samples with a negative silhouette width were removed from further analysis Next, we use either the PAMR[45] algorithm, a machine-learning based method, or a generalized linear model[46] to identify characteristic upregulated or downregulated genes of each subgroup. In a last step, functional enrichment will be performed based on a Gene Set Enrichment Analysis[47]. Predefined genesets such as given by the MSigDB[48] will be used. The PAMR-score or the predictive-score extracted from the generalized linear model was used to identify subclass specific gene set enrichment. Pipeline and codes are available at https://github.com/heilandd/AutoPipe and CRAN AutoPipe.

**Functional analysis by enrichment analysis**. A permutation-based pre-ranked Gene Set Enrichment Analysis (GSEA) was applied to each module to verify its biological functions and pathways[47]. The predefined gene sets of the Molecular Signature Database v5.1 were taken. Enrichment score was calculated by the rank order of gene/metabolite computed by random forest accuracy[47]. For significant enrichment, p-values were adjusted by FDR. Gene Set Variation Analysis (GSVA) was performed with the GSVA package implemented in R-software. The analysis based on a non-parametric unsupervised approach, which transformed a classic gene matrix (gene-by-sample) into a gene set by sample matrix resulted in an enrichment score for each sample and pathway[49].

**FACS postprocessing and tSNE**. FACS data are processed by the flowCore package[50] to gate samples, remove tumor cell contaminations and extract profiles of astrocytes. Processed FACS data was used to calculate the over-dispersion of the protein intensity of each cell and constructed a cell–cell distance matrix. Data was illustrated in a two-dimensional matrix by t-Distributed Stochastic Neighbor Embedding (tSNE) as implemented in package "Rtsne" for R.

**Reporting summary**. Further information on research design is available in the Nature Research Reporting Summary linked to this article.

## Data availability

RNA-Sequencing Data available: GSE128536, Accession codes: www.github.com-/heilandd/. Further information and requests for resources, raw data, and reagents should be directed and will be fulfilled by the Contact: D. H. Heiland, dieter.henrik. heiland@uniklinik-freiburg.de. Full table of all materials is given in the supplementary information. The source data underlying Figs. 1c, e, 2b–c, 3d, f, h, 5e–h, and 6c–f are provided as a Source Data file.

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

## Acknowledgements

D.H.H. is funded by the German Cancer Society (Seeding Grand TII), Müller-Fahnenberg Stiftung and Familie Mehdorn Stiftung. We thank Jonathan Göldner for his continuous support in the lab.

## Author contributions

The study was designed and coordinated by D.H.H. Purification of astrocytes and transcriptopmic analysis was performed by S.P.B., J.W., S.H., and J.H.F. FACS analysis was performed by S.P.B., M.S., and A.L.P. Tissue purification and sampling was performed by D.D., J.G., C.F., O.S., and JB. Neuropathological analysis and IHC was performed by R.S. and M.P. MR-Imaging and neuroradiology analysis performed by I.M. Organotypic slice culture and Immunostainings performed by V.M.R., K.J., N.W.C.G. Bioinformatical analysis are performed by D.H.H. and K.J. Manuscript writing editing and figure design was performed by D.H.H., J.S., P.F. O.S., U.G.H., and J.B. helped oversee the project.

## Additional information

**Competing interests:** The authors declare no competing interests.

