## [Peer Review File · Nature Communications]

Reviewers' comments:

Reviewer #1 (Remarks to the Author):

In this study, the authors investigate whether tumor-associated astrocytes play a critical and active role in establishing an immunosuppressive tumor microenvironment through cross-talk with tumor cells and microglia. Initial characterization of astrocytes isolated from varying regions (i.e., tumor core, infiltrating margin, and distant cortex) of primary human glioblastoma revealed that tumor core-associated astrocytes have a higher propensity for proliferation (clustering near published fetal astrocyte profiles) and for anti-inflammatory reactivity (clustering near published ischemically-reactive "A2" reactive astrocytes). Additionally, as characterized by GSEA and immunohistochemical labeling, IFN γ and JAK/STAT pathways were highly upregulated in tumor-associated astrocytes. Subsequently, the authors used several functional assays (i.e., organotypic slice culture and co-culture) to more closely examine astrocytes and/or microglia activity in relation to tumor cells. In both systems, astrocytes and microglia significantly increased anti-inflammatory cytokine release (i.e., TGF β , IL10, etc.) when able to interact with microglia or astrocytes, respectively, and tumor cells. Evidence is also presented that microglia/myeloid cells isolated from regions containing astrocytes exhibited gene expression phenotypes associated with being less inflammatory, which the authors suggest that peritumoral astrocytes exhibit a particular type of phenotype that they refer to as 'A2', and the say that this phenotype of astrocyte "mediates" a transcriptional reprogramming of myeloid cells.

Though and interesting topic, and timely, unfortunately, there a numerous concerns regarding the manuscript, including substantial over-interpretation of what is essentially a collection of correlative data and circumstantial evidence. The findings largely reiterate in the context of human glioblastoma, observations that been made repeatedly by multiple laboratories in many different types of CNS disorders ranging from trauma to infection to autoimmune disease. The observations here largely represent an incremental advance.

Concerns with the manuscript include:

1. A requirement for more explicit commentary on which experiments/models used all primary human tissue and which used a combination of primary and immortalized cell lines. With regards to the latter concern, in the manuscript's current state, the reviewer needs to deeply read into the methods to gather this important information. It is critical to highlight the benefits and challenges of immortalized cell lines as a proxy for human systems.
2. Why are datasets not available on a publically-available database – e.g. GEO? As no data is provided about the purity of the cells collected it makes it impossible to reliably review the quality of these data. Deposition of such data is a requirement of this journal for submission.
3. Down regulation of astrocyte functions in tumor-associated astrocytes (first referenced on line 131): Please elaborate on what astrocyte-specific functions were down-regulated in tumor-associated astrocytes. Given that astrocytes have significant and varied roles in homeostasis, metabolism, synaptic transmission, and many other physiological roles, it would be really interesting to see if due to the tumor microenvironment, do astrocytes lose all of their baseline functions? Or is this loss-of-function more specific?
4. Expression of CD274 (lines 148-152): You note that CD274+ astrocyte expression had "strong differences among different regions." Could you please elaborate? What kinds of differences did you notice (i.e., intensity of CD274, localization, numbers of CD274+ astrocytes, etc.)?
5. Physiological human relevance of organotypic slice culture (first referenced on lines 159-160): This is a really interesting model, but I think additional commentary on model specifics are needed. Also, the use of human astrocyte and tumor cell lines should be explicated stated, so that readers can interpret the pros and cons of this model. Additionally, please respond to the following questions:
 - a. Why is an immortalized astrocyte cell line used in lieu of primary astrocytes derived from the slice itself? Does the chlodronate treatment affect astrocytes in the tissue (only NeuN+ neurons and IBA+ microglia are quantified and listed in immunostained sections shown)?
 - b. Is the astrocyte cell line comparable to astrocytes isolated from primary human tissue? Can this be

shown through clustering analysis of monocultures and primary astrocytes isolated from epilepsy patients, non-infiltrating zones of the cortex, or from other studies?

c. How does serum components (specifically B27) in astrocyte culture media alter astrocyte transcriptome/function in these assays? Serum is well-known to alter the transcriptomic state of astrocytes.

d. Given the novelty of model system, please elaborate on why the time courses were chosen.

6. Microglia, astrocyte, and tumor cell co-culture (first referenced on line 196): Like in the organotypic slice culture, why were immortalized cell lines used for all three cell types in this co-culture system? Please elaborate on the relevance of these cell lines in comparison to the primary cell characterization presented in Figures 1 and 2.

7. CD274 also has high neuron expression (especially cajal-retzius cells in the hippocampus (see Saunders et al 2018. Cell. 174(4) P1015-1030.E16) - how was this accounted for? How many CD274+ cells were not positive for GFAP+/S100b(/or other astrocyte marker)?

Additionally, the flow of this manuscript was challenging in parts and could be significantly improved by minor grammatical changes, so that the storyline is easily understandable by readers (please see the Specific Comments section for suggestions).

Specific minor comments:

1. The manuscript would benefit from careful reading for multiple grammatical errors and broken sentences. For example, the sentence starting on line 154 does not make sense. (other instances have not been listed here for brevity, but the manuscript should be carefully proofed before resubmission)

2. Figure 1a: gene labels for rows of the heatmap often do not align – see MTCO1P12, making these label useless.

3. Figure 4e-f: What does the acute designation stand for?

4. Line 90-94: the correct reference for A1/A2 astrocytes is Zamanian et al., 2012, J Neurosci

5. Line 83: Add citation to Zhang et al. reference.

6. Lines 132-135: Add in citations for all data sets that were compared to your own unique RNA-seq data generated.

7. Line 142: Add in citation for Zhang et al.

8. Lines 226-227: Add in citations for “age, spatial adaptation, and disease origin.”

9. Lines 250-252: Add in citations for “STAT3-activated astrocytes were found to express axon growth-promoting molecules contrary to prevailing dogma that glial scar avoid axon regeneration.”

Reviewer #2 (Remarks to the Author):

The manuscript by Heiland and co-works is a very interesting work where they identified the roles of reactive astrocytes and microglia cells in tumor microenvironment. They demonstrated that the tumor-associated reactive astrocytes contributed to the evolution of immunosuppressive environment and astrocytic activation is partially driven by tumor and microglia cells, as immunosuppressive environment is important for tumor growth. In order to address these points, the authors utilize a novel microglia chemical-depletion model in a human ex-vivo organotypic slice model with GSC injection and analyze the alteration of both microglia and astrocytes by RNA-seq based gene expression analysis. Overall, the work is well organized and presented, and it brings novel insight into the shift of gene expression of astrocytes and microglia cells in the tumor environment, which might hinder immune therapeutic efficiency.

However, there is few functional studies in this manuscript to strengthen their conclusions. In order to improve the works, below are some of the points that should be taken into consideration:

1. In Fig. 1, astrocytes from cortex specimens of epilepsy patients were used as control to identify gene expression changes in astrocytes from glioblastoma. However, it has been reported that mTOR

pathway activation in astrocytes might participate in the epileptic foci of temporal lobe epilepsy [Zeng, L.H., N.R. Rensing, and M. Wong, The mammalian target of rapamycin signaling pathway mediates epileptogenesis in a model of temporal lobe epilepsy. *J Neurosci*, 2009. 29(21): p. 6964-72.] and reactive astrocytes existed in mouse model of epilepsy. Moreover, the level of phosphorylated ribosomal protein S6, the classical target of mTOR, was induced in reactive astrocytes during the chronic phase of epileptogenesis (Wang, X., et al., Deletion of mTOR in Reactive Astrocytes Suppresses Chronic Seizures in a Mouse Model of Temporal Lobe Epilepsy. *Mol Neurobiol*, 2017. 54(1): p. 175-187). These works suggest that astrocytes from cortex specimens of epilepsy patients may bring a big concern to serve as control in the current setting. As such, the authors should make comments and convince readers that astrocytes from epilepsy specimen did not hamper the conclusions for RNA-seq results as shown in Fig.1b. Otherwise, astrocytes from paired non-infiltrating (supplement fig 1) area should be used as control.

2. The authors may evaluate the increased expression levels of CD274, STAT3-P and CH3L1 by Western blotting analysis instead of immunofluorescence alone.

3. In Fig. 2b, the authors stated as "positive astrocytes can be identified by their unique morphology". However, only morphology is not always reliable, thus objective astrocytes marker (such as HepaCAM) should be used to identify positive astrocytes as well.

4. In the manuscript, the author concluded as "astrocytic activation is partially driven by tumor and microglia cells" based on their beautiful ex-vivo organotypic human culture model. The authors claimed that chronic murine model does not reflect human situation. However, a syngeneic GL261 orthotopic xenografts mouse model in WT mice and *Csf1r*^{-/-} mice (which lack microglia), in my mind, can be used as in vivo model, which will strengthen this conclusion, because it's still not clear that whether astrocytic activation driven by microglia would have an influence on the tumor growth or not.

5. In Fig. 4a, cortex specimen of epilepsy patients was used to constructed organotypic human culture. It has been reported that reactive astrocytes exist in epilepsy tissues (de Lanerolle, N.C., T.S. Lee, and D.D. Spencer, Astrocytes and epilepsy. *Neurotherapeutics*, 2010. 7(4): p. 424-38; Pekny, M., et al., Astrocytes: a central element in neurological diseases. *Acta Neuropathol*, 2016. 131(3): p. 323-45), the author should explain whether intrinsic reactive astrocytes in organotypic human culture could influence the activation of ZsGreen tagged astrocytes.

6. Only one immunofluorescence staining is not sufficient to evaluate the expression level of protein (Figure 4d and e), I suggest to add more information, such as Western blotting.

7. The author observed strongly activation of JAK/STAT pathway in reactive astrocytes. To illustrate the significance of JAK/STAT pathway activation in reactive astrocytes, I suggest using JAK/STAT pathway inhibitors or knockdown to further evaluate the astrocytes activation and inflammatory response.

Other comments:

1. The surface marker Hepatic and Glial Cell Adhesion Molecule (HepaCAM) is highly enriched in astrocytes compared with other non-neoplastic cell populations such as OPCs, neurons and microglia cells (Zhang, Y. et al. Purification and Characterization of Progenitor and Mature Human Astrocytes Reveals Transcriptional and Functional Differences with Mouse). However, the distinct HepaCAM expression levels between glioblastoma and astrocytes were not clearly shown. The authors should comment on this.

2. It is important to describe figures in chronological order and match the sentences in the context. The authors need to rearrange the context from line 176 to 195.

Point by Point:

Reviewer #1 (Remarks to the Author):

Concerns with the manuscript include:

1. A requirement for more explicit commentary on which experiments/models used all primary human tissue and which used a combination of primary and immortalized cell lines. With regards to the latter concern, in the manuscript's current state, the reviewer needs to deeply read into the methods to gather this important information. It is critical to highlight the benefits and challenges of immortalized cell lines as a proxy for human systems.

We improved the description of each experimental setup including improved illustrations in each figure and we added a more detailed description to each figure.

2. Why are datasets not available on a publically-available database – e.g. GEO? As no data is provided about the purity of the cells collected it makes it impossible to reliably review the quality of these data. Deposition of such data is a requirement of this journal for submission.

Data now available at GEO (GSE128536). We additionally provide supplementary data showing the quality of astrocytic purification. A maximum accurate separation between tumor cells and activated astrocytes is difficult and certainly part of the limitations of the work.

We discussed: *“Although Zhang and colleagues¹³ used HepaCAM-based immunopanning to purify astrocytes from tumor specimens, currently the specificity of HepaCAM to purify tumor-associated astrocyte and the extent of tumor cell contamination remain unclear. For this purpose, we called copy number variations of purified astrocytes and tumor cells in order to prove low rates of tumor cell contamination, **Extended Data Figure 1c.**”*

3. Down regulation of astrocyte functions in tumor-associated astrocytes (first referenced on line 131): Please elaborate on what astrocyte-specific functions were down-regulated in tumor-associated astrocytes. Given that astrocytes have significant and varied roles in homeostasis, metabolism, synaptic transmission, and many other physiological roles, it would be really interesting to see if due to the tumor microenvironment, do astrocytes lose all of their baseline functions? Or is this loss-of-function more specific?

This part was removed from the current manuscript. In order to investigate the pathways that were enriched in tumor-associated astrocytes, we revise our analytic approach as shown in **Figure 1c-d**.

4. Expression of CD274 (lines 148-152): You note that CD274+ astrocyte expression had “strong differences among different regions.” Could you please elaborate? What kinds of differences did you notice (i.e., intensity of CD274, localization, numbers of CD274+ astrocytes, etc.)?

The number of CD274+ astrocytes were found to be enriched in the peritumoral region. We improved the wording of this paragraph to ensure a more accurate description.

5. Physiological human relevance of organotypic slice culture (first referenced on lines 159-160): This is a really interesting model, but I think additional commentary on model specifics are needed. Also, the use of human astrocyte and tumor cell lines should be explicated stated, so that readers can interpret the pros and cons of this model. Additionally, please respond to the following questions:

a. Why is an immortalized astrocyte cell line used in lieu of primary astrocytes derived from the slice itself? Does the clodronate treatment affect astrocytes in the tissue (only NeuN+ neurons and IBA+ microglia are quantified and listed in immunostained sections shown)?

First, our purpose was to investigate to what extent the altered microenvironment in microglia depleted slices indirectly effected the reactivity of astrocytes. Of course, the direct comparison to the astrocytes from the slices is missing. In order to improve our experimental set-up, we additionally purified astrocytes directly from the slices to guarantee a better comparison. Clodronate is not affecting astrocytes or transcriptional reactivity of astrocytes as shown in the novel transcriptional data.

b. Is the astrocyte cell line comparable to astrocytes isolated from primary human tissue? Can this be shown through clustering analysis of monocultures and primary astrocytes isolated from epilepsy patients, non-infiltrating zones of the cortex, or from other studies?

This address an important point. Cell lines, always, have the major disadvantage of only reflecting part of the actual functional diversity. In order to avoid this source of error, we aimed to restrict the number of experiments with cell lines to a minimum. In some cases, however, it is not possible to avoid an experimental setup with cell lines. Therefore, we demonstrate a comparison between primarily isolated astrocytes and cell lines (Figure 4). It was shown that with regard to the reactive status, astrocyte cell lines can be used but have limited meaningfulness. We discussed this source of error and tried to achieve a better comparability of primary isolated astrocytes and cell lines.

c. How does serum components (specifically B27) in astrocyte culture media alter astrocyte transcriptome/function in these assays? Serum is well-known to alter the transcriptomic state of astrocytes.

Slices and cells were serum-free cultured, also B27 was serum-free, a detailed information is given in the supplementary table.

d. Given the novelty of model system, please elaborate on why the time courses were chosen.

We added a section in the material and methods to explain the time course in detail.

6. Microglia, astrocyte, and tumor cell co-culture (first referenced on line 196): Like in the organotypic slice culture, why were immortalized cell lines used for all three cell types in this co-culture system? Please elaborate on the relevance of these cell lines in comparison to the primary cell characterization presented in Figures 1 and 2.

In line with the microglia-depletion model we aimed to establish an astrocytic-depletion model in order to investigate the role of astrocytes within the crosstalk between tumor, astrocytes and microglia. As expected, it turns out that astrocytic depletion is highly toxic to neurons and microglia. Further. we used

a co-culture model to study the isolated crosstalk of tumor cells and microglia as well as the changes caused by astrocytes. Therefore, we used primary tumor cell lines and immortalized astrocytes and microglia cell lines. We highlighted the limitations of this model in the discussion.

“Further, we demonstrated that distinct tumor-associated microglia activation was maintained by the crosstalk of tumor, astrocytes and microglia crosstalk. These findings were limited by the necessary use of immortalized cell lines, but concurred with the results obtained from the human slice model above.”

7. CD274 also has high neuron expression (especially cajal-retzius cells in the hippocampus (see Saunders et al 2018. Cell. 174(4) P1015-1030.E16) - how was this accounted for? How many CD274+ cells were not positive for GFAP+/S100b(/or other astrocyte marker)?

We add an improved analysis in figure 1 including GFAP (for astrocytes) and marker for immune cell landscape (CD3, IBA1, CD68, P2RY12, HLA-DR).

Additionally, the flow of this manuscript was challenging in parts and could be significantly improved by minor grammatical changes, so that the storyline is easily understandable by readers (please see the Specific Comments section for suggestions).

The manuscript was fully re-written to improve the flow and the red-line of our investigation.

Specific minor comments:

1. The manuscript would benefit from careful reading for multiple grammatical errors and broken sentences. For example, the sentence starting on line 154 does not make sense. (other instances have not been listed here for brevity, but the manuscript should be carefully proofed before resubmission)

We corrected grammatical errors and broken sentences.

2. Figure 1a: gene labels for rows of the heatmap often do not align – see MTCO1P12, making these label useless.

All figures have been revised

3. Figure 4e-f: What does the acute designation stand for?

The designation “acute” was removed. It meant: Slices that were not cultivated but were directly examined after the slicing.

4. Line 90-94: the correct reference for A1/A2 astrocytes is Zamanian et al., 2012, J Neurosci

Was changed

5. Line 83: Add citation to Zhang et al. reference.

The references were corrected.

6. Lines 132-135: Add in citations for all data sets that were compared to your own unique RNA-seq data generated.
 7. Line 142: Add in citation for Zhang et al.
 8. Lines 226-227: Add in citations for “age, spatial adaptation, and disease origin.”
 9. Lines 250-252: Add in citations for “STAT3-activated astrocytes were found to express axon growth-promoting molecules contrary to prevailing dogma that glial scar avoid axon regeneration.”
- Reply to 6-9: We added missing citations.

Reviewer #2 (Remarks to the Author):

1. In Fig. 1, astrocytes from cortex specimens of epilepsy patients were used as control to identify gene expression changes in astrocytes from glioblastoma. However, it has been reported that mTOR pathway activation in astrocytes might participate in the epileptic foci of temporal lobe epilepsy [Zeng, L.H., N.R. Rensing, and M. Wong, The mammalian target of rapamycin signaling pathway mediates epileptogenesis in a model of temporal lobe epilepsy. *J Neurosci*, 2009. 29(21): p. 6964-72.] and reactive astrocytes existed in mouse model of epilepsy. Moreover, the level of phosphorylated ribosomal protein S6, the classical target of mTOR, was induced in reactive astrocytes during the chronic phase of epileptogenesis (Wang, X., et al., Deletion of mTOR in Reactive Astrocytes Suppresses Chronic Seizures in a Mouse Model of Temporal Lobe Epilepsy. *Mol Neurobiol*, 2017. 54(1): p. 175-187). These works suggest that astrocytes from cortex specimens of epilepsy patients may bring a big concern to serve as control in the current setting. As such, the authors should make comments and convince readers that astrocytes from epilepsy specimen did not hamper the conclusions for RNA-seq results as shown in Fig.1b. Otherwise, astrocytes from paired non-infiltrating (supplement fig 1) area should be used as control.

Astrocytes were purified from entry cortex (temporal pole) of patients suffering from hippocampal sclerosis. Here we did not suspect any direct changes, but we also collected astrocytes from the entry non-infiltrated cortex (as described in extended data figure 1) of tumor patients to improve the control cohort. The new samples are now included in figure 1.

2. The authors may evaluate the increased expression levels of CD274, STAT3-P and CH3L1 by Western blotting analysis instead of immunofluorescence alone.

We added western blot and FACS analysis of primary tissue. In order to quantify the STAT3-P and KI67 in the slice model we used FACS analysis, due to the low cell number in the slice model, western blots were not possible.

3. In Fig. 2b, the authors stated as “positive astrocytes can be identified by their unique morphology”. However, only morphology is not always reliable, thus objective astrocytes marker (such as HepaCAM) should be used to identify positive astrocytes as well.

We added an improved analysis in figure 1 including GFAP (for astrocytes) and marker for immune cell landscape (CD3, IBA1, CD68, P2RY12, HLA-DR).

4. In the manuscript, the author concluded as “astrocytic activation is partially driven by tumor and microglia cells” based on their beautiful ex-vivo organotypic human culture model. The authors claimed that chronic murine model does not reflect human situation. However, a syngeneic GL261 orthotopic xenografts mouse model in WT mice and *Csf1r*^{-/-} mice (which lack microglia), in my mind, can be used as in vivo model, which will strengthen this conclusion, because it’s still not clear that whether astrocytic activation driven by microglia would have an influence on the tumor growth or not.

In this study we consciously chose a human model to compare the reactive alterations in astrocytes derived from primary tumor specimens with those in a controlled experimental set-up. The suggested murine model would add further information that are limited in our proposed model and should be considered in future studies.

The direct comparison between an orthotopic xenografts mouse model in *Csf1r*^{-/-} mice and our results is beyond the scope of this paper. We discussed the usage of our human model:

“However, the direct transfer of murine experimental data to human pathological events often fails²⁰ due to numerous differences in the architecture of the immune system²¹, astrocytes¹³ and other cell types of both species. In order to replicate human environment, we did not make use of a murine model.”

5. In Fig. 4a, cortex specimen of epilepsy patients was used to construct organotypic human culture. It has been reported that reactive astrocytes exist in epilepsy tissues (de Lanerolle, N.C., T.S. Lee, and D.D. Spencer, Astrocytes and epilepsy. *Neurotherapeutics*, 2010. 7(4): p. 424-38; Pekny, M., et al., Astrocytes: a central element in neurological diseases. *Acta Neuropathol*, 2016. 131(3): p. 323-45), the author should explain whether intrinsic reactive astrocytes in organotypic human culture could influence the activation of ZsGreen tagged astrocytes.

We thank the reviewer for these fruitful comments, we now, used slices from epilepsy as well as tumor entry cortex. We are not able to fully avoid the influence of intrinsic reactive astrocytes or reactive changes caused by slicing directly. As an additional control we validated the status of activated astrocytes in acute slices (directly after slicing) and removed samples that showed a high number of activated astrocytes (quantified by Immunostaining GFAP 32/168 slices).

6. Only one immunofluorescence staining is not sufficient to evaluate the expression level of protein (Figure 4d and e), I suggest to add more information, such as Western blotting.

We added FACS data as explained above, **Figure 4j,k**.

7. The author observed strongly activation of JAK/STAT pathway in reactive astrocytes. To illustrate the significance of JAK/STAT pathway activation in reactive astrocytes, I suggest using JAK/STAT pathway inhibitors or knockdown to further evaluate the astrocytes activation and inflammatory response.

We thank the reviewer to help us substantial improve the manuscript and results. We added additional experiments, which revealed the effect of JAK-STAT inhibition on the anti- and proinflammatory cytokine environment, **Figure 6 and last part of the results.**

Other comments:

1. The surface marker Hepatic and Glial Cell Adhesion Molecule (HepaCAM) is highly enriched in astrocytes compared with other non-neoplastic cell populations such as OPCs, neurons and microglia cells (Zhang, Y. et al. Purification and Characterization of Progenitor and Mature Human Astrocytes Reveals Transcriptional and Functional Differences with Mouse). However, the distinct HepaCAM expression levels between glioblastoma and astrocytes were not clearly shown. The authors should comment on this.

We added data to show the accuracy of HepaCAM as a marker for astrocytes. A maximum accurate separation between tumor cells and activated astrocytes is difficult and certainly part of the limitations of the work as discussed more detailed.

However, we tried to prove the purity of astrocytes in different steps. First, we called the CNV in all purified astrocytes and compared them to tumor cell lines as well as FACS analysis, which showed low co-staining of ZsGreen tagged tumor cells and HepaCAM positive astrocytes.

We discussed: *“Although Zhang and colleagues¹³ used HepaCAM-based immunopanning to purify astrocytes from tumor specimens, currently the specificity of HepaCAM to purify tumor-associated astrocyte and the extent of tumor cell contamination remain unclear. For this purpose, we called copy number variations of purified astrocytes and tumor cells in order to prove low rates of tumor cell contamination, **Extended Data Figure 1c.**”*

2. It is important to describe figures in chronological order and match the sentences in the context. The authors need to rearrange the context from line 176 to 195.

The manuscript was fully re-written and re-ordered.

Reviewer #1 (Remarks to the Author):

This revision by Heiland and colleagues maintains the interesting biological phenomena described in the first iteration of the manuscript, however address the vast majority of the concerns by this (and other) reviewer(s).

Specific comments have been added to the attached PDF, however overall the manuscript has removed over-reaching conclusions, and simplified the flow of the text.

To reiterate the key concerns of the current version of the manuscript:

POINT #2:

ORIGINAL REVIEWER COMMENT: Why are datasets not available on a publically-available database – e.g. GEO? As no data is provided about the purity of the cells collected it makes it impossible to reliably review the quality of these data. Deposition of such data is a requirement of this journal for submission.

> AUTHOR REBUTTAL: Data now available at GEO (GSE128536). We additionally provide supplementary data showing the quality of astrocytic purification. A maximum accurate separation between tumor cells and activated astrocytes is difficult and certainly part of the limitations of the work.

We discussed: "Although Zhang and colleagues¹³ used HepaCAM-based immunopanning to purify astrocytes from tumor specimens, currently the specificity of HepaCAM to purify tumor-associated astrocyte and the extent of tumor cell contamination remain unclear. For this purpose, we called copy number variations of purified astrocytes and tumor cells in order to prove low rates of tumor cell contamination, Extended Data Figure 1c."

>> ADDITIONAL REVIEWER COMMENT: This addition (and publishing requirement) is good. This reviewer is not sure that the data provided in Supp Fig 1c provides any evidence of a lack of tumor cell contamination in purified astrocytes.

POINT #7:

ORIGINAL REVIEWER COMMENT: CD274 also has high neuron expression (especially cajal-retzius cells in the hippocampus (see Saunders et al 2018. Cell. 174(4) P1015-1030.E16) - how was this accounted for? How many CD274+ cells were not positive for GFAP+/S100b(/or other astrocyte marker)?

> AUTHOR REBUTTAL: We add an improved analysis in figure 1 including GFAP (for astrocytes) and marker for immune cell landscape (CD3, IBA1, CD68, P2RY12, HLA-DR).

>> ADDITIONAL REVIEWER COMMENT: This addition does not address the point of the initial question: how many CD274+ cells were not GFAP+? That is, how many CD274 cells are not astrocytes? This quantification must be included somewhere when suggesting a new marker for a particular cell type.

The manuscript will provide key data that will be of broad interest to the scientific community. There is still some strange stylistic choices that make interpretation of the data difficult (see comments attached re: Supp Fig 6), however this version is markedly improved.

I would enjoy seeing this manuscript in press.

Reviewer #2 (Remarks to the Author):

The manuscript has been improved significantly . Their answers meet my reviews.

REVIEWERS' COMMENTS:

Reviewer #1 (Remarks to the Author):

This revision by Heiland and colleagues maintains the interesting biological phenomena described in the first iteration of the manuscript, however address the vast majority of the concerns by this (and other) reviewer(s). Specific comments have been added to the attached PDF, however overall the manuscript has removed over-reaching conclusions, and simplified the flow of the text.

To reiterate the key concerns of the current version of the manuscript:

POINT #2:

ORIGINAL REVIEWER COMMENT: Why are datasets not available on a publically-available database – e.g. GEO? As no data is provided about the purity of the cells collected it makes it impossible to reliably review the quality of these data. Deposition of such data is a requirement of this journal for submission.

> AUTHOR REBUTTAL: Data now available at GEO (GSE128536). We additionally provide supplementary data showing the quality of astrocytic purification. A maximum accurate separation between tumor cells and activated astrocytes is difficult and certainly part of the limitations of the work.

We discussed: “Although Zhang and colleagues¹³ used HepaCAM-based immunopanning to purify astrocytes from tumor specimens, currently the specificity of HepaCAM to purify tumor-associated astrocyte and the extent of tumor cell contamination remain unclear. For this purpose, we called copy number variations of purified astrocytes and tumor cells in order to prove low rates of tumor cell contamination, Extended Data Figure 1c.”

>> ADDITIONAL REVIEWER COMMENT: This addition (and publishing requirement) is good. This reviewer is not sure that the data provided in Supp Fig 1c provides any evidence of a lack of tumor cell contamination in purified astrocytes.

As mentioned in the previous comment, we cannot guarantee 100% tumor-free sample preparation. In case of a high number of tumor cell contaminations we would be able to detect the tumor specific CNVs such as the gain of Chr7 or the loss of Chr10. This was not the case with our purified astrocytes, so we assume that there is only a low level of contamination in our samples.

POINT #7:

ORIGINAL REVIEWER COMMENT: CD274 also has high neuron expression (especially cajal-retzius cells in the hippocampus (see Saunders et al 2018. Cell. 174(4) P1015-1030.E16) -

how was this accounted for? How many CD274+ cells were not positive for GFAP+/S100b(/or other astrocyte marker)?

> AUTHOR REBUTTAL: We add an improved analysis in figure 1 including GFAP (for astrocytes) and marker for immune cell landscape (CD3, IBA1, CD68, P2RY12, HLA-DR).

> > ADDITIONAL REVIEWER COMMENT: This addition does not address the point of the initial question: how many CD274+ cells were not GFAP+? That is, how many CD274 cells are not astrocytes? This quantification must be included somewhere when suggesting a new marker for a particular cell type.

The tumor-associated astrocytes are marked by CD274+ and GFAP+. As well, some macrophages (CD68 ~20%) and microglia cells (IBA1, P2RY12 ~35-40%) are also CD274+. However, in the peritumoral region 95% of the CD274 positive cells are stained positive for GFAP. Further, our reported cells show a unique morphological phenotype that is most likely to be assigned to astrocytes. Macrophages, microglia and also other immune cells (CD4/8 pos. cells) are morphologically distinguishable from star-shaped astrocytes.

The manuscript will provide key data that will be of broad interest to the scientific community. There is still some strange stylistic choices that make interpretation of the data difficult (see comments attached re: Supp Fig 6), however this version is markedly improved.

I would enjoy seeing this manuscript in press.